# Students' Perceived Priorities on Water as a Human Right, Natural Resource, and Multiple Goods

**Riikka P. Rajala** [1,*] **, Tapio S. Katko** [1] **and Gunta Springe** [2]

1   Faculty of Built Environment, Tampere University, 33720 Tampere, Finland; tapio.katko@tuni.fi
2   Faculty of Geography and Earth Science, University of Latvia, LV-1586 Riga, Latvia; gunta.springe@lu.lv
*   Correspondence: riikka.rajala@tuni.fi; Tel.: +358-50-447-9998

**Abstract:** As often noted, water is one of the most critical natural resources in the world—one we must take care of so that future generations can enjoy safe water. This study specifically explores university-level water and environmental students' views on perceived priorities on water. The recent debate on water policy and its complexity is first reviewed, followed by a study on how students perceived water through six predetermined criteria. Interactive learning events ($n$ = 241) were arranged worldwide in 2011–2015 in seven countries and one region: Finland, Latvia, South Africa, Brazil, Mexico, Sri Lanka, USA, and Southern Africa region. The relative distribution of the criteria totaling 100% were as follows: Basic human right 31%, natural resource 25%, economic good 15%, public and social good both 11%, and cultural good 7%. The views did not substantially differentiate despite the different socio-economic conditions. Yet, basic human right should be interpreted wisely remembering environmental, economic, and other realities. Here, the target group consisted of water and environmental students, and it would be very interesting to conduct a comparative study among students in other fields (sociology, economics, etc.). On the whole, we should further analyze the value of water and its priorities to make it easier to manage water resources in the future.

**Keywords:** water services; water policy; Finland; Latvia; South Africa

## 1. Introduction

A recent UNICEF and WHO report showed that globally as many as 2.1 billion people lack access to safe drinking water, 4.5 billion are without safe sanitation, and 0.9 billion practice open defecation. Furthermore, only 10 percent of global wastewater is efficiently treated. It can be argued that this lack of services is the most severe global catastrophe, and it happens silently and continuously [1]. According to McDonald et al., up to two thirds of mankind will suffer from chronic water scarcity and/or polluted water by 2050 if improvements are not made [2].

Along with these challenges, different views and perspectives seem to exist on how water should be used and managed. There are several requirements for good water management. While those can be considered justified from their own perspective, they may be contradictory to each other. Correia pointed out how "water is an essential requisite for development and an essential element of all ecosystems" [3]. At the same time, it is a subject of a complex and diversified industry. Due to its nature and importance practically in all areas of economics, water has become an object of policy debates, including many areas of social life, particularly in water-stressed areas.

According to Perry et al., there is broad support for the idea of treating water as an economic good [4]. Yet, water has several roles as a basic need, a merit good, and a social, economic, financial, and environmental resource. Therefore, it is exceptionally difficult to formulate, e.g., appropriate pricing on water. Perry et al. continue that "the flow of water through a basin is complex, and provides wide scope for externalities, market failure, and high transaction costs" [4].

Many of the Sub-Saharan countries, after gaining independence in the 1960s and 1970s, introduced a free water policy. It was considered justified in the prevailing political climate, but soon proved unrealistic. For instance, Tanzania abandoned its free water policy in the 1990s, while Zanzibar held out until 2006 [5].

Schouten and Schwartz, as well as Ioris and Costa, remind us that the political nature of the water sector and investment decisions should not be omitted [6–8]. The argument is supported by the finding that globally, approximately 90% of water utilities, 95% of wastewater undertakings, and close to 100% of stormwater systems are publicly-owned. These systems and services are characterized by an exceptionally high number of stakeholders, such as indirect and direct users, including possible industrial customers; permit and local authorities; landowners; planners and consultants; contractors; manufacturers; decision-makers; and civil servants. Using soccer analogy, D.C. North defined institutions as the "formal and informal rules of the game while organizations are the players" [9].

In international water policy, there have been interesting changes and rethinking over time on how water and water services (community water supply and wastewater) should be seen in terms of their private and public spheres. In the late 1980s, privatization of water services started to gain interest worldwide in connection with international policy changes towards New Public Management (NPM), one of them being the so-called Washington consensus [10].

In 1989, the Thatcher government carried out the most dramatic full privatization of water services of recent times in England and Wales. Before that, the Regional Water Authorities (RWAs), established in 1974, were denied of borrowing adequate funds. They got into a financially impossible situation, which led to turning the publicly controlled RWAs into private companies [11,12]. However, the main reason behind the privatization was political and ideological [13].

Soon after, linked to the upcoming United Nations Conference on Environment and Development (UNCED) in June 1992, also known as the Rio Summit, the International Conference on Water and the Environment (ICWE) was held in Dublin in January 1992, and the Dublin statement on water and sustainable development was made.

The statement included four major principles. Principle No. 1 stated that "Fresh water is a finite and vulnerable resource, essential to sustain life, development and the environment". It pointed out a holistic approach that links social and economic development with protection of natural ecosystems. The second principle emphasized that "decisions are taken at the lowest appropriate level, with full public consultation and involvement of users in the planning and implementation of water projects" [14].

The third Dublin Principle focused on women who "play a central part in the provision, management and safeguarding of water", which is likely very true in many developing economies. Finally, principle No. 4 argued that "Water has an economic value in all its competing uses and should be recognized as an economic good". However, the same principle pointed out the vital and fundamental importance of the right of all human beings to have access to clean water and sanitation at an affordable price [14].

Following the Dublin statement, the principle of economic good started to get overemphasis and, thereby, by the mid-1990s, international financial agencies started to promote privatization of water services based on long-term concessions or operational contracts. Experiments were carried out, particularly in Latin America, but also in Africa and transition economies, with contradictory results. The "privatization decade" 1995–2003, as Franceys [15] called it, proved largely a failure, as shown by international evaluations, as well as researchers like Bakker, Castro, and Vinnari and Hukka [16–22]. Although some authors find positive health impacts [23] and improved access [24] after privatization, the major concern is that excess amounts of funds seem to go to external sources. Accordingly, Ahlers states that the single-minded focus on the economic dimension relegates water valuation to its commodification [19].

A more recent and interesting trend is the return to municipal control of water services [25]. Perhaps the most well-known case occurred in the City of Paris, where 25-year operational contracts

with two multinational companies ended in 2010 [20,26]. Between 2000 and 2015, Berlin, Germany was facing a budget crisis due to losses from risk investments, and therefore decided in 1999 to sell its water utility through a complicated privatization process [27]. Yet, in 2013, the company was bought back into public ownership [26]. At the end of the day, the question of values is faced: What is it that we actually want? The point made by Wuolle as early as in 1912 in Helsinki, Finland is still very valid: "In any case, I am of the opinion that municipalities must keep control of at least the retailing and planning of supply within their borders in order to remain the master of the house and promote municipal development" [28].

Along with the privatization trials, however, water as a basic human right gradually came on the agenda. It was promoted by the United Nations Committee on Economic, Social and Cultural Rights, which, in November 2002, noted that access to water is a human right—a social and cultural good, not merely an economic commodity—and it defined the public nature of water as "a limited natural resource and a public commodity fundamental to life and health". Its history dates back to the United Nations Declaration on Human Rights in 1948. Since then, the right to better environment developed gradually, e.g., through the United Nations Conference on the Human Environment and the related Stockholm declaration in 1972, as well as the notion in the Brundtland Report Our Common Futures in 1989: "All human beings have the fundamental right to an environment adequate for their health and well-being" [29].

The process culminated in the historic United Nations resolution "Right to water and sanitation" in July 2010 [30]. Based on this, the United Nations Human Rights Council (HRC) affirmed in September 2010 that access to water and sanitation is a human right. Yet, the resolutions failed to guide us on how the costs of water supply and sanitation (WSS) can be recovered with affordable prices for the poor [31].

According to Harsha, uncontrolled urbanization and lack of planning will render any human right meaningless [32]. His view is that rivers left unmodified do not ensure human right to water and they "neither provide safe water nor deliver water to households by themselves as they are not free from microbes". Benton argues that many human social practices both serve human purposes and provide a setting for environmental value, and he argues for a re-valuation of urban open space [33].

Khadkaa says that water is legally emerging as a human right in many countries [34]. However, there are significant challenges and opportunities for implementing the idea that water is a human right. Hill notes that ethics is not all about human rights and welfare [35]. Thereby, he suggests that a proper valuation of natural environments is essential to—and not just a natural basis for—a broader human virtue that he calls "appreciation of the good". In her studies on water values and ethics, Zenner [36] mentions how many scholars agree with Bakker's [37] (p. 38) pragmatic assessment that "human rights are not the solution, but rather are a strategy for creating the context in which social and environmental justice can be pursued".

According to Barraque, there is a fear in returning water to public management by announcing a decrease in water price at the expense of asset renewal [38]. Langford argues that countries have international legal obligations to respect, protect, and fulfil the water–related human right without discrimination. The human rights approach is valuable, since it puts people first and emphasizes the need for participation, accountability, legal enforcement, and marginalized groups [39].

Yavuz states that "in spite of the fact that abundance of water creates problems and sometimes disasters, much of the problem with water lies with its scarcity". He also points out how shortages of water in a region occur mainly for two reasons: Climatic conditions, or remoteness of the water resources. Here, engineering and technology come into play, along with capital investments. Thus, project economy requires both technical and economic feasibility [40].

As Pietilä et al. point out, "water has been considered an economic as well as a public good, but is not exclusively one or the other—it is a mixture of the two and many other features" [41]. Ioris notes that the political disputes around the valuation of water are typically reduced to a simple dichotomy between economic (modern) and traditional (anti-modern) interpretations. He introduced the concept

of territorialized, positionality in which "no single value dominates completely, but multiple systems of value overlap and meaning is constantly reconstructed" [7].

Overall, it is clear that there are multiple objectives and criteria to consider for managing water and water services, some of them controversial. These include balancing the requirements between water for people, food, energy, and nature, raised since the Dublin principles from 1992. For this debate, we wish to give an additional contribution. Since students are the most educated part of the younger generation, we hope that this study could also promote global thinking in local context.

The objective of this paper is to find out how students perceive and prioritize water as a basic human right, cultural good, economic good, natural resource, public good, and social good (called criteria). The classical definition of sustainability contains technical, environmental, and social dimensions. This study produced better understanding of sustainable water services through an analysis of priorities on six dimensions and highlighted the development of global thinking in higher education in relation to the sustainability of water as a resource.

The review and objective lead us to the following research questions: (i) How do the respondents rank the selected criteria and what kind of arguments they have? (ii) How much do the results vary in the selected countries and how can they possibly be explained?

Followed by this review of the complexity of water management and services, the objective and research questions, an international survey on water as a basic human right, a natural resource, and a cultural, social, economic, and public good is presented. The questionnaire aimed at promoting students' thinking of water management globally and locally in the framework of the Dublin principles. After this results are presented, reflections made and conclusions drawn.

In the next section, materials and methods of this research and the survey with university students are described.

## 2. Materials and Methods

The survey was used as an interactive learning event in water management courses at universities, mainly at BSc and MSc levels, applying the instructions by Kapp [42]. First, a two-page slide instructional presentation was shown on the contradictory views and historical development of water policy. The first slide referred to the Dublin principles from 1992, one of them considering water as an economic good. It also mentioned The United Nations Committee on Economic, Social and Cultural Rights (UN/CESCR, Nov 2002) that pointed out the access to water as a human right; a social and cultural good, not merely an economic commodity and defined the public nature of water as "a limited natural resource and a public commodity fundamental to life and health". It further mentioned the Convention on Human Rights in 1966, and the historic resolution Right to Water and Sanitation, passed by the United Nations (UN) in July 2010.

The second introductory slide included a simplified typology illustrating the major water services options in relation to exhaustion and exclusion (Figure 1). In the case of on-site services by private systems, exclusion is feasible. In the case of cooperatives (water user associations), mainly the members are served through a club or toll good. A public good, often free of charge, may apply, e.g., to public stand posts. However, in most cases, water is a common pool resource and it is normally obligatory to connect to the systems in service areas. In developed economies, all urban dwellings have piped in-house connections and thus single use. Other authors, e.g., Cornes and Sandler [43], cited by Prevos [44] (p. 11), use the concept of rivalry instead of exhaustion. Whichever is more correct, water services come up as different types of goods, depending on their provision and scale.

After the introductory slides, a questionnaire on a paper sheet was given to participants in pairs. They were requested to discuss their views and assess the relative importance of the six key criteria mentioned above, and listed alphabetically as percentage totaling 100. It was decided not to give specific definitions on the selected criteria, but rather let students interpret these from their own perspectives. Pairs were also requested to give written arguments on why, for a maximum of two lines on the form, for each of the six criteria. At the end of the session, approximately 10 min were used for

fast compiling of the rankings and discussion. The whole session took approximately 45 min. The students were highly engaged, and thereby were expected to learn more and retain the knowledge longer, as advised by Kapp [42]. The authors found the survey interesting and it promoted lively discussions among the participants, also after the session.

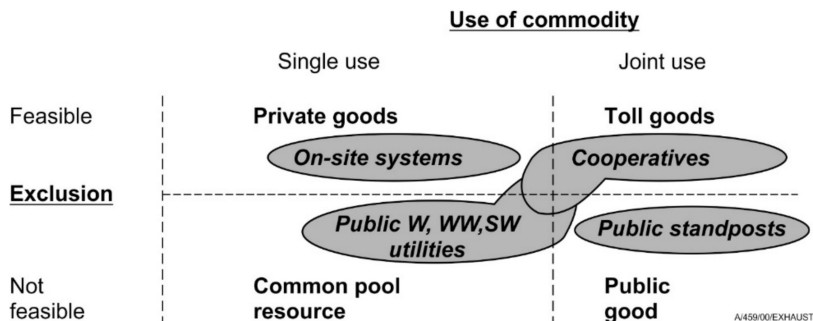

**Figure 1.** Typology of goods in terms of major water services options in relation to the feasibility of exclusion and exhaustion in terms of water use.

In the beginning, the survey was tested by a student group in Finland in 2009. Thereafter, the questionnaire survey was expanded to three groups of MSc students at Finnish universities and BSc students at universities of applied sciences. After a seminar in Newcastle in July 2011, selected foreign colleagues were gradually requested to conduct a similar questionnaire survey among their students in 2011–2015. The introductory slides were provided for this purpose by the second author. In 2011, also the third author from Latvia became involved. The authors present an inter-disciplinary team, covering environmental engineering, civil engineering, and environmental sciences.

The questionnaire was sent to professional colleagues in universities in seven countries (Table 1) and the Southern Africa region (Malawi, Namibia, Tanzania, Zimbabwe). Since the respondents were semi-professionals, mainly students in the field, they supposedly had greater knowledge and interest of the issue than an average citizen. However, they could not be considered as professionals, except for a few more experienced MSc or doctoral students. The respondents had a fairly similar background, and their views on perceptions on water represent those of young professionals in their country/region.

**Table 1.** Number of accepted respondents from each case country.

| Country/Region | Respondents | Persons Per Group | Degree Aimed at |
|---|---|---|---|
| Brazil | 17 | $2 \times 8, 1 \times 1$ | BSc in Env. Eng. |
| Finland | 31 | $3 \times 1, 2 \times 12, 1 \times 4$ | BSc and MSc. in Env. Eng. |
| Mexico | 9 | $2 \times 1, 1 \times 7$ | BSc and MSc. related to water |
| South Africa | 46 | $3 \times 2, 2 \times 20$ | BSc and MSc. in Civil & Env. Eng. |
| Sri Lanka | 31 | $3 \times 1, 2 \times 14$ | BSc in Civil Engineering |
| Latvia | 71 | $2 \times 35, 1 \times 1$ | BSc and MSc. in Env. Science |
| United States | 24 | $2 \times 10, 1 \times 4$ | BSc and MSc. in Civil & Env. Eng. |
| Southern Africa region | 8 | $1 \times 8$ | MSc. in Water Resources Eng. |
| | 237 | | |

As an exception, in Latvia, the introductory slides were translated to Latvian, while the survey itself was conducted in English. The total number of accepted respondents was 237 (Table 1), out of which the highest numbers occurred in Latvia (*n* = 71), South Africa (*n* = 46), and Finland (31). Accordingly, the results from these three countries with the largest data are shown in more detail than others (Table 1), as well as summaries of the qualitative statements (Tables 2–4). The other surveyed countries were selected to represent other varied types of regions and conditions, thus giving a more or less "global view".

**Table 2.** What kind of good is water? Summary based on open views on the six selected criteria of water in Latvia, listed according to the global ranking ($n = 37$).

| Criteria | Views from Latvia |
|---|---|
| Basic human right (23%) | Water as a basis for life and living (20)<br>Equality (13)<br>Means for survival (8) |
| Natural resource (31%) | Life on Earth (26)<br>Basis for all other water-connected aspects (9)<br>Main constituent of Earth, water cycle, important for planet existence (7)<br>Limited resource and unequal access (5)<br>Latvia is rich with waters (2) |
| Economic good (15%) | Water as a basis for economic development and human benefits (26)<br>Development of Latvian economy is strongly based on water resources (3)<br>Economical value allows to prevent excessive water consumption (3)<br>Water must be free for everyone (3)<br>Water as ecosystem service (2) |
| Public good (9%) | Equality and human right (20)<br>The role of clean water (6)<br>The role in recreation (6)<br>Water is not a good, it is invaluable, use without taxes, not private property (4)<br>Water as privilege (4)<br>Availability in Latvia (2) |
| Social good (12%) | Importance for human health (quality of drinking water, recreation) (19)<br>Regional and property differences (not important in Latvia) (9)<br>Not a basic need (4)<br>Water protection and environmental safety (2)<br>Increasing role due to increasing population (2) |
| Cultural good (9%) | Cultural and religious aspects are less important than basic human needs (12)<br>Cultural heritage with emphasis to Latvia (10)<br>Esthetical, recreational value of water landscape (9)<br>Historical role in religion, moral values and symbol (7)<br>Source of inspiration for artists (5)<br>The role in rise of civilization (2) |

**Table 3.** What kind of good is water? Summary based on open views on the six selected criteria of water in South Africa, listed according to the global ranking ($n = 23$).

| Criteria | Views from South Africa (SA) |
|---|---|
| Basic human right (35%) | Cannot live without water, necessity, essential for survival (17)<br>Needed for basic purposes; should be accessible to everyone, regardless of their financial standing, to avoid inequality; everyone has the right to survive and have proper hygiene (5)<br>High unemployment rate in SA; therefore, many people cannot afford water payments (1) |
| Cultural good (6%) | In favour:<br>Respect of cultures and traditions (5)<br>Not much water is required and varies from community to community (2)<br>Culture for the future generations (1); water for cultural practices can be from the same water supply for public, social, and basic human right (1); people of all cultures require water, but some use water more than others (1)<br>Less in favour:<br>There is no link between culture and water; not a necessity and can be used minimally (3)<br>Urbanization reduces the number of people who need water for cultural reasons (1); all people are not religious and need water for baptism (1); culture has some importance, but can be seen as a want, not a need (1) |
| Economic good (19%) | Needed for agriculture, industrial, mining, construction, etc. (8)<br>People need water, regardless of their economic situation; this ensures quality for social, public, and human right (5)<br>Infrastructure for providing water, maintenance (2) |

**Table 3.** *Cont.*

| Criteria | Views from South Africa (SA) |
|---|---|
| Natural resource (14%) | SA has vast natural resources, unique environment aquatic life and ecosystems that need to be sustained and protected (6)<br>Water as a natural attraction, other than esthetic appeal, can promote tourism, which can lead to job creation (5)<br>To preserve all natural resources, e.g., birds, animals, trees for future generations, and keeping the natural habitat alive (4)<br>SA is a dry country, therefore water is needed for the environment, needed to support the environment → economy (3)<br>Important for the environment but not a priority; very little of the natural water is visually pleasing and therefore should not take priority (2) |
| Public good (13%) | It is important to have water available and for the general public to have access, to prevent environmental degradation and improve public image (10)<br>Necessary to improve quality of life; people can have a better quality of life and wellness is enhanced, needed for high standard of living (3)<br>Community taps are important in providing water where household water pipes are not available; taps are needed in parks and malls, and supplying the need of the community, e.g., in university (2)<br>Because the public pays tax, the public should have a fair amount of water (2) |
| Social good (13%) | SA has a lot of poor, sick, and vulnerable people that need help (11)<br>People need access to water in households, community centers, and social homes (2) |

Latvia represents a country that experienced Restoration of Independence in 1990, and the third author is a Latvian citizen. In South Africa, the Apartheid legislation was abolished in 1991, and multiracial elections were held in 1994. In addition, all South African citizens are constitutionally guaranteed access to a certain amount of water free of charge [45]. The first author is also familiar with the South African conditions. Finland is the home country of two of the authors, representing a state where water services developed fast after World War Two and reached a high level of service according to several international water- and environmental-related comparisons.

Although a comparative approach is used in the study, we wished to see how these identified six key criteria are perceived in different socio-economic and cultural conditions. The survey was expanded gradually to five invited teachers in different countries and regions to see if the general pattern from the initial three countries would prove different in other conditions or not. The sample of the selected countries is no doubt limited. Yet, considering the small resources available and the voluntary contribution of the invited partners, it can be seen as reasonable.

The data analysis included calculated average values for priorities of perceptions on water in case countries, the box and whiskers graph for the whole data, comparative results in Latvia, South Africa, and Finland, as well as overall results from seven countries and one region.

In the next section, the results are first introduced based on global view, and then more specifically in three case countries: Latvia, South Africa, and Finland. Finally, a short comparison of all case countries is made. While exploring the results, it is good to keep in mind that the educational and other background of each student, as well as the school books used, will affect their views and perceptions. For instance, by using a critical geopolitics perspective, Ide et al. [46] noted that German textbooks seem to overstate the risk of water conflicts. This is in line with the findings by Rosling et al. [47]. In their global survey, they found that people, experts, and even world leaders seem to have a far too negative perception on the development that has taken place, especially in developing economies after World War Two. Accordingly, the representation of water scarcity in Jordanian textbooks was studied by Hussein [48]. He pointed out how discourses are key in constructing people's understanding of issues.

**Table 4.** What kind of good is water? Summary based on open views on the six selected criteria of water in Finland, listed according to the global ranking (*n* = 17).

| Criteria | Views from Finland |
|---|---|
| Basic human right (38%) | Water is essential, everyone needs it, without water humans will not survive, basic for life (11)<br>Everyone has to have access to water, equality (2)<br>Everyone should have right to water, not related to their economics (1)<br>Health reasons (1) |
| Natural resource (27%) | We are not alone in this world; health, biodiversity, humans are part of nature (4)<br>Water resources should be safe to use (3)<br>Not infinite, a limited natural resource (2)<br>Protecting resources; subject to pollution (2)<br>Part of sustainable development (1) |
| Economic good (10%) | Important for business, industry, agriculture, industry, hydropower (4)<br>Creates jobs (2)<br>Income to the state (2)<br>Economical profit should not be important; water should not be considered as an economic good (3)<br>Water treatment (1)<br>Price of water should not be the most important meter, enough to guarantee good/reliable service (openness, transparency) (1)<br>To be able to keep the systems operational (1) |
| Public good (10%) | All equal with water; everyone should have equal access (2)<br>Everyone needs for living (1)<br>Any area to cover its water needs (1)<br>Water maybe distributed better by non-profit organizations (2)<br>Also, in Finland, water is a common public good, but it is probably not possible worldwide (1) |
| Social good (10%) | Everyone should have equal access to water resources; jobs (3)<br>In Finland, water is a common social good, but probably not possible worldwide (1); not private (1)<br>Requires collaboration—no-one left aside (1)<br>Water is needed in leisure too (1)<br>Everyone needs it for living (1)<br>Conflicts, refreshing (1) |
| Cultural good (6%) | Historical perspective—cultures and societies grow around waterways, close to water. Finnish landscape and history include a lot of water; for Finns, water, especially lakes, hold a cultural value (2)<br>Important part of the nature (2)<br>High importance is recognized; importance for mental health (2)<br>In Finland, everyone is used to having water and accessing it freely (1)<br>Cultural aspects are not so important than other aspects (2) |

## 3. Results

### 3.1. Global View

First, the results based on the whole data are shown. As seen in Figures 1 and 2, the basic human right was considered the first priority of water according to responders (x = 31%). Natural resource was second (25%) and economic good third (15%). Public and social goods were quite even (11%), while cultural resource was a less mentioned priority (7%). The dispersion of the figures is shown in Figure 3. Since the criteria used are not independent, more detailed statistical analysis could not be seen justified.

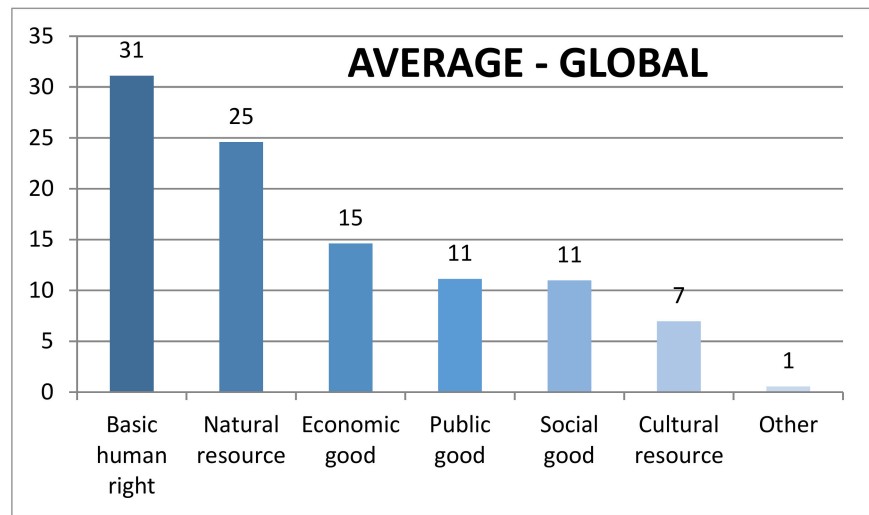

**Figure 2.** Priorities of perceptions on water. How would you divide 100% between the given perceptions on water? This graph shows answers in declining order according to the global average. The dispersion is shown in Figure 3. Number of forms $n = 131$, total number of respondents $n = 241$.

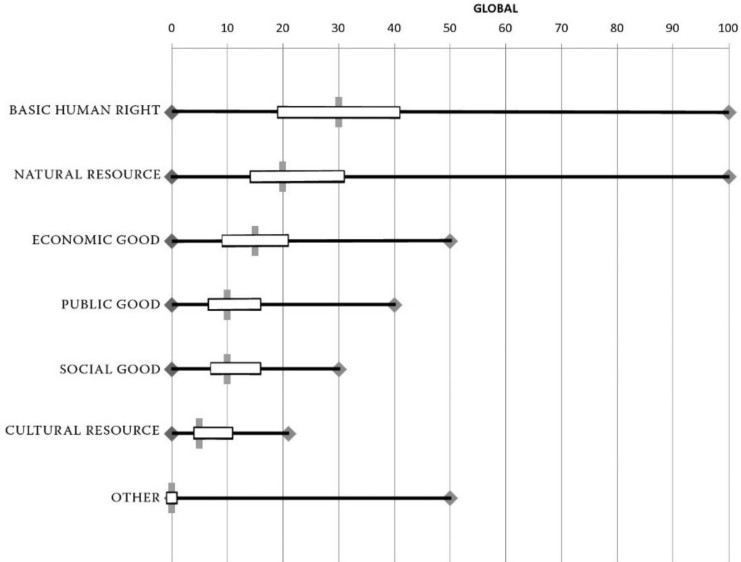

**Figure 3.** Water use priorities. How would you share 100% between the given water usage? The box and whiskers graph (median value, 25 and 75% fractiles, minimum and maximum) shows the dispersion of the given percentages. Forms $n = 131$, respondents $n = 241$.

*3.2. Case Countries: Latvia, South Africa, and Finland*

3.2.1. Latvia

As seen in Figure 4, most Latvian respondents (31%, $n = 71$) primarily considered water as a natural resource. This is not surprising, as Latvia is one of the richest states in the world in terms of fresh water resources per population, exceeding current and expected future demands [49]. The main arguments are that water is the main constituent of Earth's surface, and any existence of life is not possible without water. Water as a natural resource is thus called a keystone for all other values.

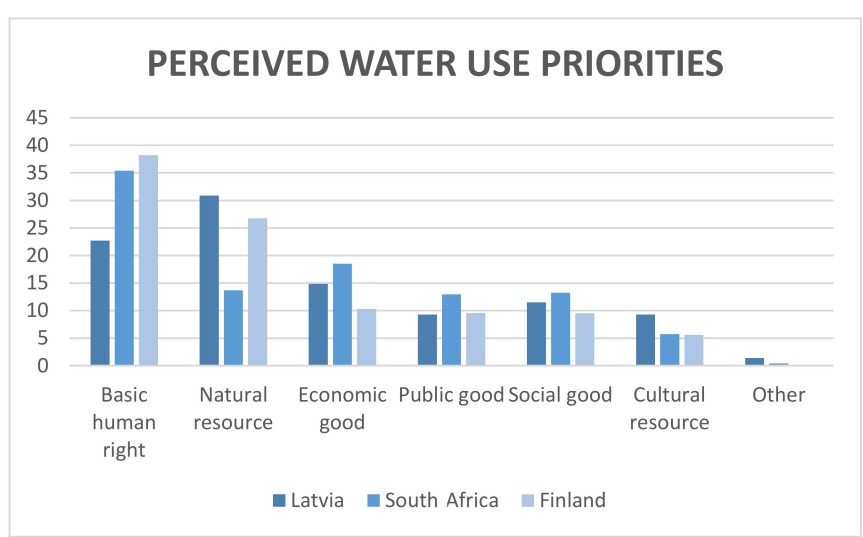

**Figure 4.** Perceived water use priorities in Latvia (no. of groups = 37), South Africa (*n* = 23), and Finland (*n* = 17), according to the global ranking.

The next category (23%) in Latvia saw water as a basic human right, mainly due to its necessity for life and living. Thus, global thinking based on individual human needs is evident. Regarding the value of water, one respondent stated: "Without water there will be no life. Water is a base of normal and productive human subsistence. Humans only have one thing that is most precious—their health. But if enough clean and healthy water does not exist, other necessary fields for meaningful living will not be as much as equivalent as with water".

Water as a natural resource is vital for life and linked with equal rights. The arguments to provide water of good quality are represented by such a principle as polluter pays. This agrees with consumer views in Cyprus and Latvia, that stated that "the analysis of consumer attitudes in the two case studies suggests that when water supply is unreliable, reliability takes precedence; once it is reliable, quality issues come to the fore" [50]. Besides, water as the main constituent of the human body is very important for survival. Accordingly, the access of water demand for clean water is put in the foreground. However, equity is found to be the next most important aspect after necessity for life and living and understanding of solidarity and responsibility is evident.

No economic development is possible without water use, which concerns also access to safe drinking water. On the one hand, economic development is connected to the increase in manufacturing and human well-being, and on the other hand, to water for human settlements. Reflecting public good, water was viewed through its quality, and importance of access to water at all levels of society worldwide interlinked with human rights.

Some Latvian respondents viewed that water is not a good at all, since it is invaluable. In the category of social good, respondents pointed out the water quality for human health. Water is also fundamental for civilization, highly valued as a cultural heritage and water landscape, including our responsibility for the next generations. Yet, the role of water as cultural good was relatively more important historically. Qualitative, written statements from Latvia are summarized in Table 2.

### 3.2.2. South Africa

In South Africa, the first priority of water was basic human right (35%), while economic good (19%) was mentioned before natural resource (14%) (Figure 4). South Africa is a country located at the southern tip of Africa, with a population of 57 million (2018). The results are understandable, since in 2008, approximately 5 million South Africans lacked access to water and 15 million lacked access to basic sanitation. As more people migrate into cities from rural villages, the pressure for the cities to meet the water demands is ever increasing, such as the case in Cape Town in 2018 [51]. South Africa

boasts one of the cleanest water systems in the world. Yet, due to lack of sanitation and access to water in the country's rural communities, the threat of waterborne disease is steadily increasing [52].

Written arguments behind the economic good were, for example, these: "People pay for clean, good water in urban areas while in rural they do not—people need to pay for water, because good and safe water is required, expensive infrastructure and without paying for water people will waste it".

Written arguments behind the water as public good included these: "Community taps are important in providing water where household water pipes are not available. People do not pay for parks, its free and the good for public to have water in the place where they spend time".

Views on water as a cultural good in South Africa were somewhat controversial, as some of the respondents favored it more than others. Qualitative, written statements from South Africa are summarized in Table 3.

### 3.2.3. Finland

In Finland, two criteria were highlighted more than others: Basic human right (38%) was the first ranked category, while natural resource (27%) was the second. The next ones—economic good, public good, and social good—were rated fairly evenly, gaining approximately 10% each. The relatively low rate for water as an economic good could be due to the fact that the major water users, forest industries, have their own systems. It was also stated that: "Price of water should not be the most important meter, enough to guarantee good/reliable service (openness, transparency)".

The rating made by Finns is easy to understand, because the country has been said to have endless water resources. Finland is rich in surface waters, almost a tenth of the country's land area is covered by water. Groundwater is widely used by local residents and by waterworks, since it is often much purer and better protected from contamination than lake and river waters. Water as a cultural good was seen as follows: Historical perspective—cultures and societies grow around waterways, close to water. Finnish landscape and history include a lot of water, for Finns water, especially lakes, hold a cultural value". A summary of the qualitative statements from Finland is shown in Table 4.

Concerning basic human rights, water is essential for life. Three respondents pointed out that everyone should have the right to clean water. Also, health issues and thirst were mentioned. As for cultural goods, especially Finnish lakes were mentioned: There is plenty of drinking water available, but also, many lakes are open to public with free entrance. It was mentioned that water is important in landscape, and it was seen as positive for mental health.

It was also mentioned that water should not be an economic good and water should not be used for business and profit. On the other hand, it is understandable that industry uses water, and water is also used in agriculture and energy production. Water utilities employ people, and those owned by cities are sources of income for cities, while managing the systems will bring costs to water suppliers.

As for natural resources, it was considered important to protect clean water resources and also protect nature's diversity. It was stated that everyone should have access to water. Water was seen as a public good rather than private. Water resources are not located evenly, though.

### 3.3. Comparisons of Case Countries

In the three focus countries, Latvia, South Africa, and Finland, the same three criteria on water were ranked highest, although their order was slightly different. Latvia ranked water as a natural resource first, while South Africa, Finland, and all the other case countries prioritized the basic human right option. The high ranking as a human right in Finland was somewhat surprising, whereas for South Africa, it was more expected.

The overall results from the seven countries and one region resemble each other remarkably (Figure 5). The six criteria used on water were ranked in almost all cases in this order: Basic human right, natural resource, economic good, public good, social good, and cultural good.

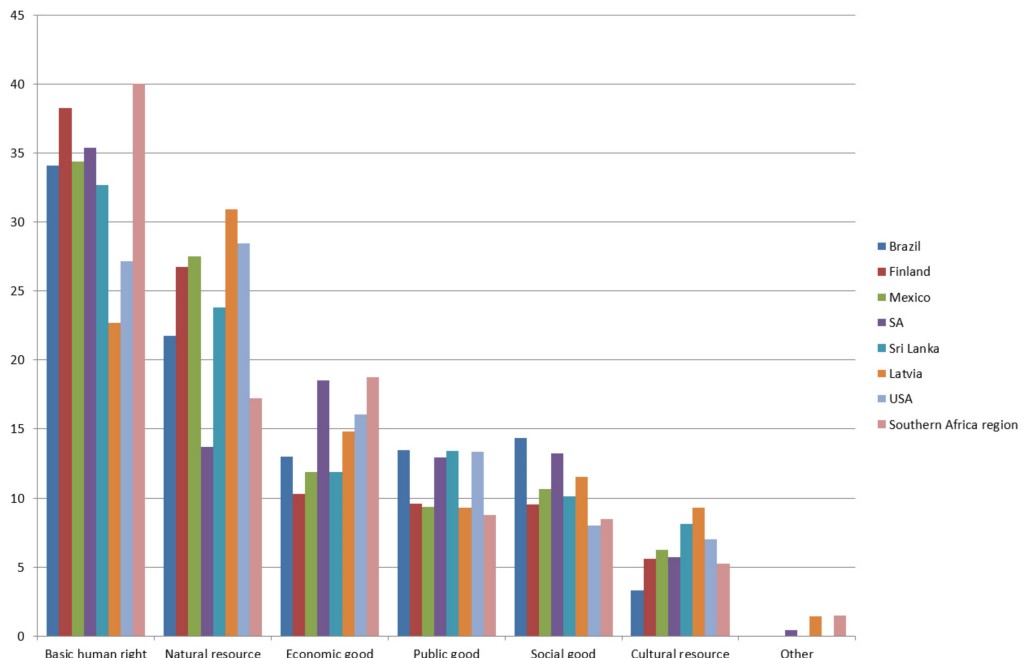

**Figure 5.** Overall results from seven countries and one region, compiling global ranking on perceived priorities.

Since water is vital to life, it is likely not a surprise that water as a basic human right is deemed the most important priority. In Latvia, several respondents linked natural resources with human rights. It was interesting to notice that people largely recognize water as an economic good, while also, critical views on its privatization were expressed. When services are operational, they are paid and properly managed. Also, the question was raised whether water should be free to some users or not.

Water as a cultural good or resource had relatively the smallest, yet not a forgotten role. Water surely has many roles in different cultures—Finns use a lot of water in saunas, many religions use water in baptism and other holy activities, and there are many beautiful water fountains in cities for softening the built environment. However, the difference between public good, social good, and cultural good may be flyaway in some cases. For example, in South Africa, public goods are close to basic human rights through access to services.

Through their open-ended written statements, the participating students explained to the authors how they understood the six selected criteria. Thus, the students actually cataloged their views under the criteria. Under each criterion, there were so many answers that the authors had to "group them" and combine similar statements. This interpretation by the authors was undoubtedly subjective and may produce some biases. The numbers of "grouped" views are presented in the respective tables. However, we need to acknowledge the qualitative nature of this survey. Since students in various countries with different conditions made their interpretations bottom-up, their views are undoubtedly subjective. Thus, we should interpret the results with care, more as suggestive rather than any exact figures.

In the next section, the results are further discussed and selected views by the invited teachers are introduced.

## 4. Discussion

The learning experience of the study is reflected here by the views of three of the invited teachers in Africa, Asia, and the USA. Professor D.A. Mashauri from Namibia says that the students are aware of knowledge that is passed from one generation to another. There were regulations, rules, or norms on how water was managed even prior to the colonial rule. The centrality of water has become proximate

to conflicts and even wars, e.g., in Golan Heights, Euphrates, and the Tigris; the Nile; Lake Malawi; and the Okavango delta. Water is indeed a universal element that can cause conflicts [53].

Professor S.B. Weerakoon states that Sri Lanka has a long history of a hydraulic civilization, with water resources management given great importance, while ancient technology is found to be sustainable and given more attention now. It has an even longer tradition of culture, in which water was regarded as a purifier and life giver, a destroyer of evil, and a symbol of fertility and mystery. However, the water right is a serious issue in the country today, and thus, the survey was found useful and interesting to teacher and students who have inherited the passion to be involved with water-related discussions [54].

In Colorado, USA, students seemed to lack awareness of the full range of benefits and goods provided by different aspects of water resources management. The survey proved to be a useful tool to stimulate their thinking and heighten their awareness of the importance of water [55].

Regarding the research questions, the results indicate that views from the various countries are largely along the same lines, and there is no substantial difference between the various socio-economic conditions. It is, e.g., likely that the students are more familiar with the concept of basic human rights as public goods. Besides, differentiation between the six criteria of this study was not always clear, since many of them are interlinked and may seem overlapping. Water as a social good, especially in South Africa, is to help the poor and alleviate poverty, whereas in Finland, this is taken care of by social security services, not by water utilities. Thus, this methodological ambiguity is openly noted.

The study provided somewhat similar findings with our previous survey that explored priorities of 10 water use purposes in 11 countries/regions on 5 continents: As they are, and as they should be [56]. Out of the 10 water use purposes in average (as they should be), community water supply was ranked first, nature conservation second, and hydropower third. Yet, the explored water use priorities varied less than anticipated. Similarly, International Law Association pointed out how water and wastewater services fulfil the "vital human needs" of communities, and thereby play a fundamental role in societal and community development [57].

Grigg pointed out that although water is a separate sector, it is also a connector in society [58]. Kishimoto et al. identified that, from 2000 to 2015, altogether, 235 cases of private water utilities were returned to municipalities in 37 countries [59]. More recently, in 2018, UK Environment Secretary Michael Gove launched a withering attack on high pay and dividends in the water industry [60]. Biswas reminds us about the value of water and points out that in many respects, the problem is not water scarcity, but its mismanagement [61].

The examples above indicate that private profit making is largely in contradiction with the fundamental ideas of public services, as well as water as a basic human right. According to Prevos, in terms of house connections, water utilities normally charge for private goods, whereas for safeguarding health, water services belong more to the public service sphere [44] (pp. 11–13). In his book, *Precious Commodity*, Melosi explored the long-term public versus private water management in the USA and pointed out how they are intertwined [62]. Han et al. remind that water utilities need to make a balance between two distinctive managerial goals, cost-effective water services and customer satisfaction [63].

Our survey implied that the debate on public or private good is partly biased and too straight-forward. Private goods and the private sector have their own roles, while the public sector fundamentally takes on the major responsibilities through appropriate rules of the game, i.e., institutions. According to Compagnucci and Spigarelli, "Water-related issues are becoming a limiting factor for sustainable economic growth and require a collaborative and interdisciplinary approach" [64].

Our study and feedback from the invited teachers imply that teachers are likely motivated to adopt or develop new educational innovations stemming from self-efficacy, as also found out by Habib and Deshotel [65]. The survey proved useful in teaching, created a lot of discussion, and after the actual survey, it was later used in teaching, at least in Finland, Latvia, and Namibia. During this study, the idea of expanding it by including such issues as drinking water, health, water transport, mental aspect, and spiritual value was born, directing the authors for a further study.

In the next section, final conclusions of this research are drawn.

## 5. Conclusions

Water has several dimensions as a natural resource, and it is also a critical resource for human life. That is why the value of water and its priorities should be analyzed for making it easier to manage in the future.

Today´s water and environmental students will be obliged to solve increasingly complex water issues in the future, and likely also issues that we are not yet able to foresee. Therefore, it is important to create dialogue on the priorities of water use purposes. Thus, it is important to raise different views also to wider discourses and awareness. If we are able to solve even one problem through increased awareness, the study has been useful. It seems that the views of students participating in our survey did not remarkably differ from each other. On the other hand, some differences occurred due to various conditions of the case countries.

It is concluded that while water and especially water services management rely and depend on local conditions, our study on perceived priorities of water implies a more general pattern and logic. The six criteria used in the survey were perceived in the following order: Basic human right as the highest, followed by natural resource and economic good, public and social good equally, and cultural good as the lowest. Yet, the differences in priorities in various countries proved to be lower than expected.

As a wider policy, we would like to recommend that the highest ranked basic human right should be interpreted wisely, preferably as the public responsibility of arranging access to services. Yet, water in environment, economic realities, as well as water as public, social, and cultural goods, they all need to be considered in a balanced way. While billions of people suffer from inadequate water services, it is important to understand how the various criteria of the study could be better taken into account. We hope that this study will, in its part, lead to productive dialogue in the future and, accordingly, the survey could be made also with students in other disciplines.

**Author Contributions:** Conceptualization, R.P.R. and T.S.K.; investigation, R.P.R. and T.S.K.; writing—original draft preparation, R.P.R.; writing—review and editing, R.P.R., T.S.K., and G.S.; visualization, R.P.R.

**Funding:** This research was funded by the VEPATUKI research cluster at Tampere University, and the Academy of Finland, under Grant [number 288153].

**Acknowledgments:** The following experts kindly cooperated in organizing the survey at their universities: N.S. Grigg (USA), Johannes Haarhoff and Johann Tempelhoff (South Africa), Leo Heller (Brazil), Blanca Jiménez-Cisneros (Mexico), D.A. Mashauri (Southern Africa region), and S.B. Weerakoon (Sri Lanka). The authors thank all of the above people, and Laura Inha, as well as peer reviewers and editors, for valuable comments and feedback.

**Conflicts of Interest:** The authors declare no conflict of interest. The funders had no role in the design of the study; in the collection, analyses, or interpretation of data; in the writing of the manuscript, or in the decision to publish the results.

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
