# Peer review of "Students’ Perceived Priorities on Water as a Human Right, Natural Resource, and Multiple Goods"

_sustainability, doi:10.3390/su11226354_

Round 1

Reviewer 1 Report

The paper is very interesting. The perception of the value of water is interesting for the future governance of the resource as it's a critical resource for human life but also it has different dimensions that should be careful analyzed to ease the management of the resource and in that regard authors do a good job analyzing how students (related to water) understand the different dimensions of water as a resource but also as a "commodity".

In relation to the paper I should recommend some important reviews.

Abstract should be redone. It seems more a telegram than an abstract. 

The introduction should be revised. First of all, some paragraphs are unconnected one from the other. That is, the authors finish explaining something and start explaining another idea without connecting both or, at least, explaining to the readers what's the link between them (for example, the privatization of Thatcher in line 56 and then, without any connection, beyond the years, explain the Dublin statement in 1992. There is no link between the privatization and then the Dublin principles. There are more paragraphs in the introduction without link)

Also, in introduction, authors should help the readers to follow the next steps of the paper once reading the introduction, that is, I should recommend authors to indicate, explicitly the next sections of the paper before starting section 2.

About methods I have some important doubts. First of all, authors decided not to give definitions for the criterias used but let students interpret them. So, if the criteria is subjective from each student in each country, is it possible to compare results? That is, the "idea" of a public good is similar in a well developed country with a good background of public utilities like Finland with the "idea" of public good in South Africa where 20% of rural population has not access to safe water? That is, students from countries could be biased to answer one criteria instead of another not for the specific criteria but maybe due to characteristics of their countries. In addition, without specific criterias can we believe that the idea of cultural resource is the same in Latvia or in South Africa? So, is it comparable what a public good is in South Africa with what is a public good in Finland?

In addition, what's the difference between public good and social good? Is it easy to distinguish, both of them? In my opinion, it can be misunderstood, normally by applied students, what are the differences between a public good and a social good. Not only that but, in my opinion, the use of public good it's not correct. I will explain it later.

Finally, I have some doubts about some bias of the study, and authors should specify it very clear in their paper. The authors claim that they are explaining the priorities of students on water, but, in reality, they are explaining the perception of very well informed students in water sector on water, that is, they are doing it's "experiment" with students related with water studies, only applied by the way, and I think it's an important bias. Although it's understandable the limitation due to economic costs, concluding that these results apply to students perception should be wrong. In addition, for me it has been strange not to include social sciences in the research, if water is an human right and depends on human beings, why not include students of social sciences. Moreover, if one of the categories of water is economic, not including students of economics is biasing the results. That is, I think that with a more heterogeneous group of students results should be different. If we include students of political sciences results of public or social good could increase, but if we include students of economics, the result of economic good could increase... The group of students used in the paper, and its background, enough representative for confirming that the paper shows the priorities of students?

About results. It has been a good idea to clearly indicate what are the answers for each criteria for each country, but they should, at least, justify it. For example, why the role of clean water indicates that it is a public good and the importance for human health a social one? Or why "people need water regardless of their economic situation" is a criteria for economic good but "it is important to have water available and for the general public to have access" a public good in South Africa? The answers are too similar to catalog in two different ones... Finally, why include an answer saying specifically "water should not be considered as an economic good" in the criteria of economic good (Finland). Is it contradictory?

Also, it should be included an explanation about how authors catalog each answer in each criteria.

Finally, the conclusion part does not explain, properly, what's the importance of the study, what authors have done, the main results and lessons learnt with this paper and some specific policy recommendations from the results. Conclusion should be wider.

Minor comments:

There is a contradiction in the first paragraph. McDonald et al said in 2011 that 2/3 of mankind will suffer from water scarcity but nowadays only 10% of the population has not access to safe water (UNICEF and WHO 2017).

About the policy of free water, authors claim that it was a failure in Sub-Saharan countries but only explain two cases. That is, these cases are representative of a majority of scenarios or only specific ones? 

Authors claim about the failure of water privatizations in Latin America but they do not explain the reasons behind (economic or political reasons) and also it's not what some authors have found (for example Galiani et al. 2005 found that water privatization in Argentina improved health in children; Clarke et al. 2009 show that privatized utilities and public utilities in Latin America improved access in similar ways, so neither public or private ones did worst in comparison with the other). Justification of their claim should be interesting.

WSS is introduced as an acronym in line 104 but, if I am not wrong, the acronym it's not explained here neither previously.

Authors claim that 90% of water utilities are public and then say "These systems and services are characterized by exceptionally high number of stakeholders"... but as far as I know, if a utility is publicly owned the stakeholder is the Government, so, if utilities are nor partially privatized, they do not have any other stakeholder... so who are these high number of stakeholders that authors say? (line 47-48)

In order to compare countries, a mean comparison test (t test) could be used to confirm if differences of perceptions between countries are similar or different.

Reviewer 2 Report

A well written paper with a focus on water and perceptions of students It would be nice if they can make the objectives of the research more explicit - maybe it will be good if they can mention why students? Their selection criteria of the students belonging to BSc or MSc - in South Africa they interview civil engineering students in addition to Env engineering where as in Latvia and Finland it is Env eng and env science - maybe. Comparison of how the engineering and science students perceive water will also be interesting as these are the three focus countries in the paper Why a focus on engineering students and not also considering social sciences? It would be nice if the authors can provide us with a bit more details on how water is perceived by law and the policies in the three selected countries to provide a basic understanding to the readier of water in these countries It is striking that this paper does not mention the last two most relevant studies in the field; this paper I believe should clearly and openly mention that it aims at contributing to this literature, building on the following articles: 1)    Ide, T. (2016). Critical geopolitics and school textbooks: The case of environment-conflict links in Germany. Political Geography, 55, 60-71. 2)   Hussein, H. (2018). A critique of water scarcity discourses in educational policy and textbooks in Jordan. The journal of environmental education, 49(3), 260-271. 3)    Ide, T., Alwan, A., Bader, K., Dougui, N., Husseini, M., Imad, E., ... & Spielhaus, R. (2018). The geopolitics of environmental education: An analysis of school textbooks in the MENA region. Journal of Educational Media, Memory, and Society, 10(2), 64-83.

Round 2

Reviewer 1 Report

The revised paper is a better one in comparison with previous one, in general the new version is more elaborated. The abstract explains better what authors did and find, also the introduction is easier to follow. Moreover the authors explain what they do in the paper in a more easy way and also explain the bias of the study, and that's even more important than showing only the results.

For the newer version I have some comments.

The main question I have with the study is the comparison between countries. That is, some of the results of the paper are that the views in different countries are similar but that statement is based on subjective answers of the students of each country that depends on their subjective point of view. That is, the idea of public, social or economic good can be very different in different countries so it's a bit strong to say that the views are similar because students in different countries grouped the answers in categories that are subjective to its own views. That is, the perception of public good is different in different countries and also similar answers in different countries are grouped in different categories. For example the answer "people need water regardless of their economic situation..." is not very different to the answer "everyone should have equal access to water resources" but the first one is categorized as economic good in SA and the second one in social good in Finland. So, the direct comparison of aggregation of categories between countries, for me, seems strange. We are including similar answers in different categories in different countries or, on the contrary, different answers in similar categories in different countries. The country explanation is really good and interesting, but I am not sure if the country comparation is so interesting due to the bias of the subjectivity of answers in each country... If authors want to do the aggregation of results or the comparation between countries should explain better what's the utility of including it or explain better that results are influenced by country's subjectivity. 

That is, authors must answer this question, as results and terms are defined by students, so results are subjective, is realiable the comparison between countries?

Apart from that, two minor comments:

About UNICEF/WHO and McDonald, first of all, apologize for my mistake, the 10% people wothout access included "basic service" as defined by WHO/UNICEF, taking into account safely managed water, the numbers of people without access increase to a 29% (in 2015). But, from 1990 to nowadays the number of people with access has increased due to the effort applied. So, if access has increased in last 25 years and efforts have been done, and nowadays 7 out of 10 people have access to safe managed water (and 9 up to 10 to basic service) I don't see why it's relevant to say that in 30 years we will be reversing numbers, that is, instead of 7 out of 10 having access only 3 out of 10 will have access... not only I do not see the relevance, but also for me it's totally contradictory because it's based in a study from 2011 supposing no effort on water access and the reality, in 2015, was that the effort not only was done but also numbers of people without access is decreasing. 

Second minor comment, about explaining what authors are going to do in next chapters seems that was misunderstoond, sorry if I did not explain it clear, when I said "In introduction, authors should help the readers to follow the next steps of the paper once reading the introduction, that is, I should recommend authors to indicate, explicitly the next sections of the paper before starting section 2", I thought about explaining just at the end of the introduction the structure of the paper. In that way, any reader reading the paper will know what authors are going to do in the next steps of the paper so they will have a clear picture of the structure of the paper and, for example, they can easy find the parts of the paper they are interested in. It's only my opinion. 

Reviewer 2 Report

I can see that the manuscript has improved.

Author Response

Thank you. On the whole, we highly appreciate your thorough review and constructive comments.